# Immunization strategies in networks with missing data

**Samuel F. Rosenblatt**[1,2]*, **Jeffrey A. Smith**[3], **G. Robin Gauthier**[3], **Laurent Hébert-Dufresne**[1,2]

**1** Department of Computer Science, University of Vermont, Burlington, Vermont, United States of America,
**2** Vermont Complex Systems Center, University of Vermont, Burlington, Vermont, United States of America,
**3** Department of Sociology, University of Nebraska-Lincoln, Lincoln, Nebraska, United States of America

* Samuel.F.Rosenblatt@uvm.edu

**Data Availability Statement:** All relevant data are available at https://github.com/sfrosenb/sfrosenb-Immunization_Strategies_in_Networks_with_Missing_Data.

## Abstract

Network-based intervention strategies can be effective and cost-efficient approaches to curtailing harmful contagions in myriad settings. As studied, these strategies are often impractical to implement, as they typically assume complete knowledge of the network structure, which is unusual in practice. In this paper, we investigate how different immunization strategies perform under realistic conditions—where the strategies are informed by partially-observed network data. Our results suggest that global immunization strategies, like degree immunization, are optimal in most cases; the exception is at very high levels of missing data, where stochastic strategies, like acquaintance immunization, begin to outstrip them in minimizing outbreaks. Stochastic strategies are more robust in some cases due to the different ways in which they can be affected by missing data. In fact, one of our proposed variants of acquaintance immunization leverages a logistically-realistic ongoing survey-intervention process as a form of targeted data-recovery to improve with increasing levels of missing data. These results support the effectiveness of targeted immunization as a general practice. They also highlight the risks of considering networks as idealized mathematical objects: overestimating the accuracy of network data and foregoing the rewards of additional inquiry.

## Author summary

It is often useful to track how epidemics spread through populations by mapping transmissions between people, communities, and cities. This consideration of a population as a network can reveal the critical players, locations, or events driving epidemics. Similarly, by mapping the network of possible transmissions before an outbreak occurs, we can identify potentially critical actors on which public health interventions should focus. Unfortunately, the data collection process required to map all possible interactions of a population is difficult—fraught with possible error and unlikely to be complete. To understand the role of data quality in network-based interventions, we apply different strategies to partially-observed networks with controllable amounts of missing data. Our results suggest that intervention strategies which require full network information remain fairly effective up to high levels of missing data. However, local strategies which rely only

**Funding:** G.R.G. & J.A.S. are supported by the National Institute of General Medical Sciences of the National Institutes of Health (Grant No. P20 GM130461) and the Rural Drug Addiction Research Center at the University of Nebraska-Lincoln. L.H.-D. is supported by the National Institutes of Health 1P20 GM125498-01 Centers of Biomedical Research Excellence Award. S.F.R. is supported as a Fellow of the National Science Foundation under NRT award DGE-1735316. The funders had no role in study design, data collection and analysis, decision to publish, or preparation of the manuscript.

**Competing interests:** The authors have declared that no competing interests exist.

on small data samples can outperform the more data-expensive benchmark strategies when little data is available. Surprisingly, we also propose an intervention that improves in effectiveness with less data by coupling targeted immunization with targeted data recovery. These results show that insights from network science can be robust to missing data, but that their implementation should be adjusted for noisy real-world applications.

## Introduction

The spread of infectious diseases [1], computer viruses [2], and "fake news" [3] pose serious threats to the health and well-being of an increasingly connected society. Thus, one of the most important questions in public health and network science is how to curtail contagions. If dangerous contagious pathogens or content spread over network connections, what kinds of interventions will be most effective at inhibiting outbreak size [4–9]? For example, prior work has shown that when the population's contact structure can be modeled as a complex network, immunization (broadly defined) of certain actors can prevent contagion considerably more effectively than randomized immunization [10–21]. With targeted immunization, actors are immunized based on their potential role in future outbreaks, which is determined by their position in a contact network [22, 23]. More applied work has demonstrated the utility of network based intervention [24]. For example, researchers have leveraged network properties to maximize the impact of their interventions in a diverse set of contexts including smoking interventions in schools [25], HIV spread among men who have sex with men (MSM) [26], and disease spread in needle sharing networks [27].

Here, the term "immunization" refers to anything that reduces the probability of infection to zero for a particular actor for the entire duration of a particular "contagion". In practice, this could be a vaccine, but other interventions, such as rehabilitation or pre-exposure prophylaxis, could also conceivably permanently reduce risk of infection at near 100% effectiveness. Targeted immunization is particularly attractive in cases where resources, like vaccines, are limited, as only a small proportion of the population must be immunized to effectively reduce wider contagion [28].

An extensive literature on networks and immunizations has focused on evaluating selection strategies [11, 29, 30]. The basic question is how to pick which actors in the network should be treated. A common choice is to evaluate immunization strategies using simulation (as we do in this paper), where a researcher stochastically spreads an infection through the treated network, recording the outbreak size under different strategies of node-immunization. The goal is to identify immunization strategies that will reduce the spread of infection the most, given the number of actors that can be treated (with limited time, money and so on).

One potential complication to identifying important actors is missing data. Many of the network-based immunization strategies suggested by existing research rely on global network measures, like betweenness and degree centrality, which assume that a researcher can map out the full network of relations on the population of interest. This is often not realized in practice, as studies are often subject to missing data [31, 32]. Missing data will potentially alter the measurement of the network [33, 34], and consequently, the identification of important actors [12, 13].

This article addresses the effectiveness of different targeted immunization strategies under conditions of missing data, see Refs. [35] and [36] for recent calls to address related problems. This is a pressing problem as some of the best immunization strategies in fully-observed networks [12, 14, 16] are based on measures that are sensitive to missing data [33, 34, 37]. For

example, past work has found that measures like betweenness, which are dependent on the full network structure, are often badly biased when nodes are missing [33, 38]. Thus, we may think that betweenness will be effective at finding important actors to immunize when missing data is low, but may not fare so well when levels of missing data are high. Alternatively, a strategy that is less effective in a true (completely observed) network may be robust to missing data, making it a potentially attractive option when missing data is a known problem. A strategy is effective to the extent that it reduces outbreak size at a given level of missing data. A strategy is robust to the extent that its effectiveness is not reduced with increasing levels of missing data. Thus, we suggest that most researchers 'on the ground' face a tradeoff between effectiveness and robustness. This tradeoff has been largely downplayed by past work, which has focused on the effectiveness of different options under the assumption of complete (or full) network information. In this article, we address the tradeoff between robustness and effectiveness directly by examining the robustness of different immunization strategies to varying levels of missing data.

We begin the article by discussing the existing research on targeted immunization strategies. We then turn to a simulation-based test of immunization strategies under conditions of missing data.

## Network-based immunization strategies

Immunization strategies work by identifying key nodes and immunizing them against infection (thereby preventing them from infecting others). Different strategies use different criteria to select the nodes to immunize. The question is which strategies are most effective in terms of preventing, or mitigating outbreaks of contagion on a population.

Two of the most commonly evaluated immunization strategies are degree and betweenness immunization. With degree immunization, nodes are immunized if they have many connections, or high degree centrality, and thus a higher chance of spreading the infection to their neighbors [11, 12, 14, 15]. Betweenness immunization selects actors for removal based on how crucial the actor is in connecting different communities (or groups), directly measuring how many shortest transmission chains can be broken via immunization [12, 14, 17].

Other studies have explored the effectiveness of stochastic approaches to identifying important actors in the network. Stochastic immunization algorithms rely on the local network, or neighborhood, of sampled nodes, rather than on global network data [18, 39]. One commonly tested stochastic strategy is acquaintance immunization, which immunizes those nodes frequently found to be neighbors of randomly selected nodes [14, 18, 40]. This strategy locates high degree nodes by relying on the proportional relationship between the degree of a node and the probability that it will be the randomly selected neighbor of a randomly selected node [41]. Acquaintance immunization is thus useful as it allows the researcher to find hubs in the network without a costly and sometimes infeasible network census.

## The problem of missing data in choosing an immunization strategy

Past tests of these immunization strategies have almost uniformly assumed ideal conditions, where there is no missing data. In the case of global network measures like betweenness, this means having information on all nodes and all ties between nodes. In the case of stochastic processes, like acquaintance immunization, this means having a complete sampling frame and complete local network information for each sampled node. However, in many cases it will be difficult to obtain complete network data. Missing data can arise for a number of reasons. For example, a researcher may not have sufficient time or resources to find and gather information about everyone in the network, particularly if the network is large. Even respondents that are

interviewed may offer incomplete information about their social contacts, being prone to for-getfulness and fatigue [31, 42, 43]. These issues are amplified in hidden, difficult to reach populations, such as a network of drug users or sex workers [44–50].

Missing data may have important consequences for choosing an immunization strategy. First, missing nodes will generally be unavailable for immunization, even if they could be identified (which would be difficult for most strategies). This is a problem endemic to network intervention studies and is built in to our exploration of the effect of missing data on immunization effectiveness. Second, missing data can affect the measurement of network properties, like degree and betweenness [34]. The rank order of nodes (from least to most important) may deviate from the true rank order on the complete, but unknown, network [51, 52]. Thus, among the set of nodes that can be immunized (i.e., those who actually participate) one runs the risk of picking a sub-optimal target set to be immunized.

We see similar issues with the stochastic strategies, like acquaintance immunization. While stochastic strategies do not require global network data, and are thus often touted as robust to missing data [14, 19, 40], implementing these algorithms in practice still requires a comprehensive list of nodes from which to sample, as well as perfect local information about the neighborhood of each sampled node. Thus, critical nodes that would be selected by these local strategies can still be overlooked either because they are missing from the local information of sampled nodes (and thus undiscoverable) or because some of their neighborhood is missing and they are thus less likely to be identified as critical compared to nodes whose neighborhoods are more heavily represented. Therefore, even a "robust" strategy, like acquaintance immunization is potentially affected by missing data.

These two ways that missing data may appear need not occur together. In certain data collection scenarios, it may be be feasible to obtain a complete sampling frame and accurate local information; in others, it may be possible to obtain one of these but not the other, and in some scenarios it may impossible to obtain complete data for either one. We will explore the effectiveness (in terms of reduction in outbreak size) of this approach under different assumptions about data availability.

## Accounting for missing data in immunization strategies

A limited number of studies have considered missing data problems related to network contagion and immunization. Most of these studies have not tackled the problem of targeted immunization problems directly, however. Hébert-Dufresne et al. [12], for example, investigate the effectiveness of immunization strategies and their robustness separately, testing effectiveness under the assumption of perfect data, and using the Jaccard coefficient to test robustness of targeting. Gong et al. [20] address how the effectiveness of an immunization strategy is affected by random network changes, but still assumes perfect data about the modified network when calculating immunization targets.

Another line of work examines the effect of missing data on objectives and metrics related, but not equivalent to, targeted immunization. These include attack robustness of networks [53, 54], influence maximization [55], and identification of influential spreaders [56]. See Ref. [57] for a discussion on the differences between influence maximization and targeted immunization.

The two studies that are most closely related to our own both investigate the effectiveness of targeted immunization strategies under specific sampling schemes. The first is a study by Yang et al. [21] who examine the potential of immunization strategies in the case of Fixed Choice Design (FCD) sampling. Fixed Choice Design puts limits on the number of neighbors a node can have, creating bias in the observed network. Measurement error of this sort can affect the

targeting strategies, but only in a limited way, as all nodes in the network are still assumed to be present and available for immunization. In this way, Yang et al. capture the effect of missing edges but not missing nodes, the focus of this study.

Chen and Lu [13] similarly consider immunization strategies under a particular sampling scheme, Respondent Driven Sampling (RDS) [47]. With RDS, a set of initial seeds are recruited into the study; these initial seeds recruit others into the study, who recruit others into the study and so on. Chen and Lu develop an algorithm to identify which actors should be immunized under such sampling conditions. The analysis is limited, however, to data collected via RDS, which does not cover the more general problem of missing network data; additionally, they assume rather ideal conditions of the RDS chain, where there is wide coverage of the population and no actors refuse to participate.

In sum, past work has focused on specific sampling schemes, while relying on the complete portion of the data to originate from an accurate process free from complications. In this article, we move past previous research by directly examining the effectiveness of targeted immunization strategies in reducing outbreak size under conditions of missing data. We allow the missing data to exist anywhere in the network, mimicking the scenario where pieces of the network are unobserved as actors are unwilling or unable to participate in the study—rather than consider data based on well-behaved sampling schemes. We also extend past work by considering the mechanisms under which stochastic strategies are subject to missing data. More substantively, we focus on the effectiveness/robustness tradeoff of different immunization strategies, making it possible to see under what conditions different strategies will be most effective at reducing outbreak size.

We now turn to testing the robustness and effectiveness of different immunization strategies. We consider a range of missing data scenarios, from the ideal case of complete information to more difficult cases where much of the network is not available.

## Materials and methods

There are six basic steps to testing the robustness of different immunization strategies: 1) select a known, true network as a test case; 2) remove nodes from the network to simulate conditions of missing data; 3) take partially-observed networks (from step 2) and identify nodes to immunize under different strategies; 4) run multiple outbreak simulations through the true network (from step 1) with selected nodes immunized (from step 3); 5) repeat steps 1-4 many times for each condition of interest; 6) compare the resulting estimates of outbreak size under each immunization strategy and level of missing data.

### Select known network as test case

Following past work on epidemics [39, 58], we use synthetic networks to test the validity of different immunization strategies. We consider synthetic (generated) networks with similar properties as the well known Colorado Springs high risk network [59]. The population of interest in the Colorado Springs network included at-risk individuals for HIV and HCV transmission, including drug injectors and sex workers. Researchers attempted to identify the entire at-risk population in the city and include them in their study. Ties are defined across several relationships: including close friendship, sexual contact, and drug co-use [59]. Disease risk networks of persons who inject drugs are of particular interest for several reasons. The sharing of injection drug apparatus is one of the most prevalent transmission routes for HIV both in terms of transmission rate [60] and also in terms of incidence rate among persons who inject drugs (PWID) [61]. Therefore, the threat to life caused by outbreaks within these populations is greater than for many other types of outbreaks in other populations. Additionally, PWID are

known to be a hidden population [47, 62, 63], thus the missing data treatment approach we present here is of special interest in regards to immunizations within these populations.

In addition, we replicate our analyses using synthetic school networks. We generated these school networks to have properties that match the friendship network structure of a typical school collected as part of the well known National Longitudinal Study of Adolescent to Adult Health (Add Health) [64]. Ties are based on self-reported friendship nominations that were collected in the classroom [65]. The Add Health network provides a case where the pathogen of interest requires close social contact, like drink sharing or hand holding; this would fit infections like streptococcus or mononucleosis.

We use synthetic networks to control relevant features, while systematically varying key network properties. From the Colorado Springs data, we generate three sets of networks sharing some basic features (e.g., composition, degree distribution, strength of homophily on key attributes like gender and race), but each with different levels of transitivity between sets. Transitivity is defined as the proportion of two-paths ($i \rightarrow j \rightarrow k$) that also have a direct path ($i \rightarrow k$). Transitivity is negatively related to contagion potential [66, 67] and the Colorado Springs network we set out to emulate has particularly high transitivity. We generate five thousand "high" transitivity networks, each with transitivity constrained around 0.30, close to the true Colorado Springs network, five thousand "medium" transitivity networks, with transitivity constrained around 0.155, and five thousand "low" transitivity networks with transitivity constrained around 0.01.

Synthetic networks are generated using exponential random graph models (ERGM) [68]. All networks are undirected, unweighted, and have 1000 nodes. In each case, the degree distribution is pulled from the original network, plotted in Fig 1A. For each of the three network types ("low", "medium" and "high" transitivity), we generate 5000 networks from the underlying model. In the S1 Appendix section 1, we replicate our analysis with other classes of synthetic networks selected from the targeted immunization literature, to demonstrate comparable results and exemplify the generality of our methodology.

## Remove nodes to simulate conditions of missing data

To simulate conditions of missing data, we followed the standard procedure in the literature: for each true network, $G$ (five thousand per transitivity level), we removed a portion of the nodes at random to form the observed network, $G'$. The removed nodes were not present in the observed network [37, 38]. The observed network ($G'$) is the network from the point of view of the researcher and serves as input into the immunization strategies.

## Identify nodes to immunize under different strategies

In general, an immunization algorithm works by identifying the nodes which are most central to the network or have a particular structural position which is considered to be important for the flow of contagion. The nodes identified with the highest scores will be the ones targeted for immunization in the subsequent outbreak simulations (with ties broken randomly). We considered eight different immunization strategies. There were three global strategies—degree, self-reported degree and betweenness—and one stochastic strategy, acquaintance immunization, was used in four different variants. We also considered random immunization as a baseline, where nodes were randomly selected to be immunized. In addition, we also run simulations with no immunization, to provide another point of comparison.

**Random immunization.**   Random immunization selects nodes at random from the given node list of $G'$, which, from the perspective of an interventionist, should be considered the sampling frame. Since nodes are missing from $G'$ at random, and random immunization

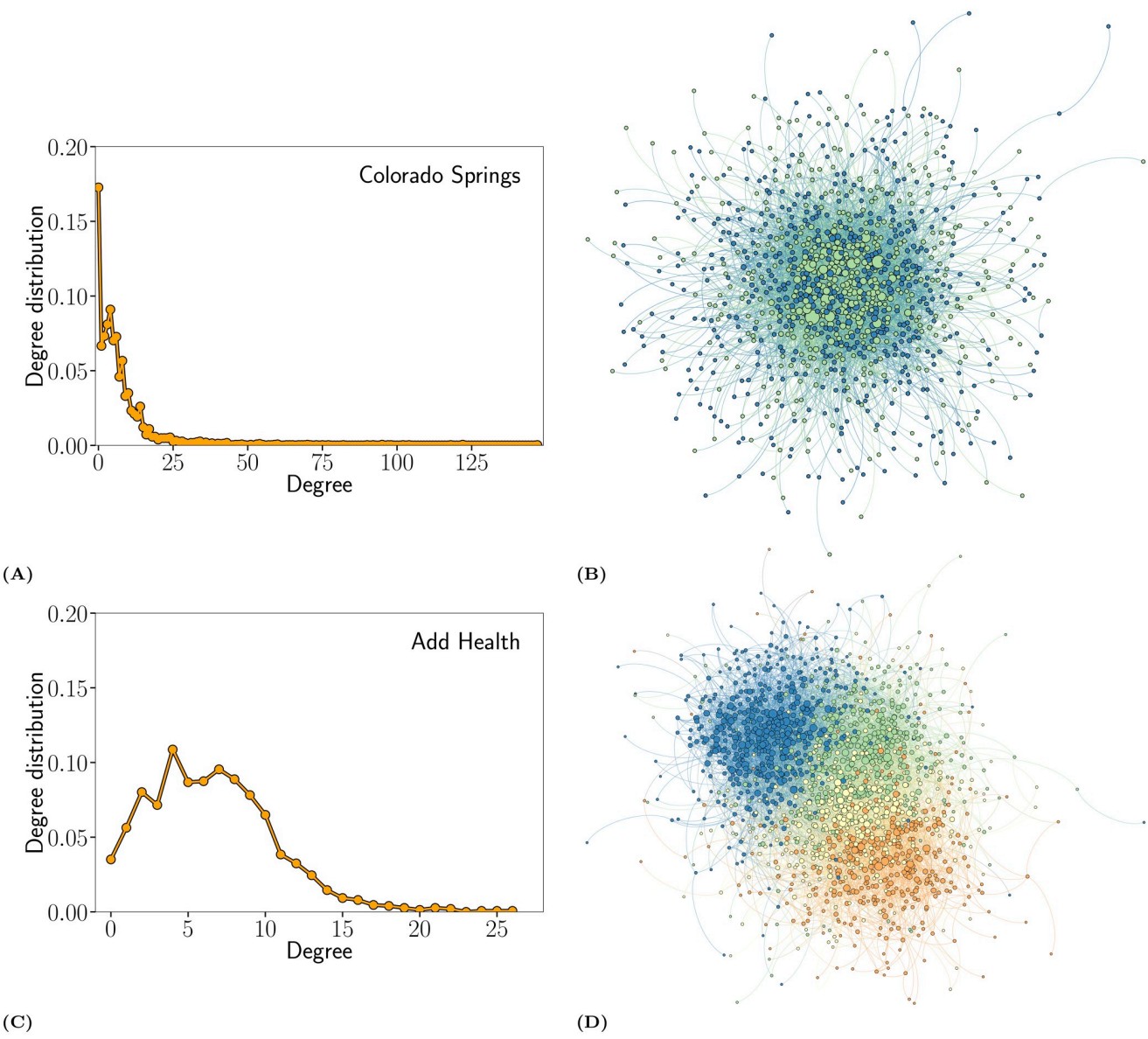

**Fig 1. Degree Distribution and Network Structures Based on the Colorado Springs (A-B) and Add Health (C-D) Networks.** In the visualization, a node's color corresponds to some additional individual attribute—gender for the Colorado Springs network and school grade for the Add Health network —while a node's size is fixed by their degree. The Colorado Springs network is relatively well described through its heterogeneous degree distributions with some additional structure (e.g. clustering) due in part to gender. The Add Health network has a more homogeneous degree distribution, but a clear modular structure that emerges since connections are more likely within than across school grade. Note that the degree distributions are based on the two empirically observed networks, one for Colorado Springs and one for Add Health.

selects a random selection of nodes to immunize from the random sample of observed nodes, random immunization of nodes from $G'$ is equivalent to random immunization of nodes from the true network, $G$. Note that this would not be the case with a non-random sample of observed nodes.

**Degree immunization.** Degree immunization selects nodes based on their network degree in $G'$—the number of ties each node receives/sends according to observed network data.

**Betweenness immunization.**    Betweenness immunization targets nodes with the highest betweenness centrality in $G'$. Betweenness centrality is the proportion of the shortest paths between nodes in the network (geodesics) upon which a node lies (counting as fractions of paths when there are multiple shortest paths between the same pairs of nodes) [69, 70]. This can be formally written as:

$$BC(v) = \sum_{\{u,w \, \epsilon \, V \, | \, u \neq v, w \neq v\}} \frac{\sigma_{u,v,w}}{\sigma_{u,w}} \qquad (1)$$

where $v$ is an arbitrary node, $\{u, w \, \epsilon \, V \, | \, u \neq v, w \neq v\}$ is the set of all pairs of nodes $u, w$ where neither $u$ nor $w$ equals $v$, $\sigma_{u,w}$ is the total number of shortest paths between $u$ and $w$, and $\sigma_{u,v,\,w}$ is the number of those paths which pass through $v$.

**Self-reported degree immunization.**    Self-reported degree is an obvious strategy that is frequently overlooked, yet it can often be straightforwardly implemented and can directly tackle missing data in degree immunization. Self-reported degree immunization targets nodes in the sample (the node list of $G'$) based on their true degree (their degree in $G$), assuming nodes know their neighbors and faithfully report their number of connections. This describes a scenario, where by survey or other query, a limited amount of accurate egocentric data on a sampled subset of nodes can be collected. This is a common assumption among most social network studies, including those which explicitly acknowledge the difficulty of obtaining a network census and thus do not presuppose complete network data, such as RDS studies [13, 71, 72] and egocentric studies [73, 74]. With complete data, this strategy is perfectly equivalent to the degree immunization described above. With missing data, we do not have access to the nodes outside of the sampling frame (and thus cannot immunize them) but missing nodes are still counted as connections by their neighbors (as they would be on a personal survey). Note also that access to this information does not require any contact with nor information about the sampled nodes' neighbors other than their existence.

**Acquaintance immunization.**    We used acquaintance immunization to test stochastic immunization strategies. We offered a test with four different versions of acquaintance immunization, with different assumptions about the information available to the researcher. Each variant makes different assumptions about the completeness of the sampling frame ($G$ for true graph and $G'$ for known graph), as well as the level of local information (again, $G$ for true graph and $G'$ for known graph).

Acquaintance immunization operates under the "friendship paradox", the principle that if one randomly samples the network, nodes that are highly connected are more likely to be neighbors of randomly sampled individuals [18, 75]. Consequently, to identify important nodes, the algorithm randomly samples a node, $r$, then randomly samples a neighbor of $r$, node $n$, and adds one to the acquaintance score of $n$. If this score increment causes $n$'s acquaintance score to go above some chosen threshold (usually 1 or 2), then n is added to the list of structurally important nodes to immunize. This process is iterated until $v$% of nodes have been identified as structurally important, where $v$ is the desired immunization level [40]. Following previous research [14], we used a threshold of 2 for adding nodes to the list of nodes to immunize.

**($G$, $G$) Variant.**    We defined the ($G$, $G$) version of acquaintance immunization as the case where both the node set for sampling and the local neighborhood information for each sampled node come from the true network, $G$. The ($G$, $G$) version of the algorithm is directly comparable to those used in existing literature without consideration of missing data. The ($G$, $G$) version is applicable in certain limited cases where missing data is a problem for global strategies, when there is neither the time nor resources to conduct collect fully saturated network

data, but a complete sampling frame exists and a limited number of surveys can be conducted accurately, such as in a school or a small government with an accurate census. Regardless, including this variant allows us to compare the best-case scenario of stochastic strategies to the global strategies.

**$(G', G')$ Variant.** In the $(G', G')$ variant, both the randomly sampled nodes and their local neighborhood information come from the observed, incomplete network, $G'$ (not the true network, $G$). Thus, only nodes in the observed network, $G'$, are available to be sampled and are listed as neighbors of the sampled nodes. This also means that only nodes in the observed network are able to be immunized. The $(G', G')$ variant is the most comparable to the global algorithms, as the process only depends on information from the observed network data. More generally, we see the $(G', G')$ variant as the most realistic version of acquaintance immunization, especially with hidden populations.

**$(G', G)$ Variant.** The $(G', G)$ version tests the scenario where no complete sampling frame exists, but perfect local information is attainable. Thus, the node set for a $(G', G)$ version is incomplete, but the local information about the sampled nodes' neighborhood comes from the true network, $G$, which has complete information. This means that the researcher is able to identify and immunize any neighbor of nodes in the sampling frame, even if that node is not in the observed network, $G'$ (meaning no refusal of treatment). This scenario exemplifies an ideal, theoretical case of using acquaintance immunization for hidden or hard to reach populations. For instance, settings where the neighbors for each node can be uniquely identified (even if they are not initially in the study, e.g. with a phone number), where the sampled nodes' trust, memory, and knowledge does not restrict information about their neighbors, and where the locatability and trust of the sampled nodes' neighbors does not restrict the ability to treat those neighbors (neither through absence nor refusal).

**$(\bar{G}, G)$ Variant.** The $(\bar{G}, G)$ variant is similar in concept to the $(G', G)$ version above and assumes that the researcher's sampling frame is initially incomplete, but since their network information is valid it can be used to supplement the sampling frame. For example, a researcher following an acquaintance immunization scheme is initially limited to the incomplete sampling frame when selecting a random node, but that node (who is undergoing a surveying process already) can inform the researchers of their neighbors. In the $(\bar{G}, G)$ variant, we allowed nodes queried in the process to add their neighbors to the sampling frame for the subsequent discoveries and immunizations. This variant therefore leverages the immunization procedure to continue data collection and continuously uncover missing data (if any).

## Immunization and SIR infection model

The next step in the test took each version of the true network, $G$ (from step 1), immunized the nodes identified by each strategy for that version of $G$ (from step 3) and simulated epidemic dynamics through the treated (static) network. We considered 3 immunization levels: 5%, 10%, and 15%. For example, with the 10% immunization level, 100 people in a network with 1000 nodes were immunized, determined by the rank ordering of important actors specific to that immunization strategy (if global), or the selection of important nodes through random exploration (if stochastic). This immunization is assumed to be 100% effective on the single contagion being modeled.

Once the important actors were immunized in their own copies of the original network, we used Monte Carlo calculations of the Susceptible-Infectious-Recovered (SIR) model to estimate the final outbreak size for each immunization strategy (for a particular parameter set) by running multiple groups of simple SIR simulations through the treated networks. In SIR simulations, every infected node infect their susceptible neighbours at rate $\beta$ and recover at rate $\gamma$.

The SIR model runs until there are no longer any infected nodes. We assume the infection runs its course without further intervention. This model was chosen for its simplicity and applicability to varied scenarios.

For our simulations, we made use of the exceptionally fast implementation by St-Onge et al. [76]. For simplicity, we set time units to be equal to the expected recovery time (such that $\gamma = 1$) and vary $\beta$ over a large enough range to explore outbreaks of varied sizes. In addition to these parameters, we also report the corresponding $R_0$ values defined as the average number of secondary cases by an infectious individual in an otherwise susceptible population (the simulation software [76] uses the average number of secondary cases caused by the second infectious individual).

More generally, the goal of our analysis is to examine the effect of missing data on immunization strategies for a generalized outbreak model. Thus, our results are not based on a single infection (e.g., HIV) but instead, we look at a range of cases where each case could be thought of as a different type of infection, with its own infectiousness and recovery rate. The epidemiological case could thus be anything transmitted over network connections. The input parameters in the simulation are consistent with a range of infections, such as HCV and HIV (lower infection potential) to varicella or mumps (higher infection potential).

One common approach in the targeted-immunization literature is to consider only large-scale outbreaks during this simulation process [12, 14]. This method avoids issues related to the frequently bimodal nature of SIR outbreak distributions, see Ref. [77] and section 2.2 of the S1 Appendix, but fails close to the epidemic threshold when there is no systematic threshold to be used. Instead, we were able to reach a tight confidence bound on the mean outbreak size for each set of parameters we tested, despite the extreme variability of many outbreak size distributions, by using $10^7$ Monte Carlo simulations for each parameter set. We spread these $10^7$ simulations out over 5000 separate networks that we generated with similar characteristics to reduce the risk of spurious results that can occur from only testing on a single network or a small number of individual networks. Each of these 5000 networks was sampled to simulate missing data before removing nodes for each strategy and for every immunization level; 2000 SIR simulations were then run for each of these treated networks and the mean of these 2000 simulations was recorded and placed in a distribution of means over all 5000 networks. The values we report are the grand mean of these 5000 means. We separately recorded the size of each individual outbreak and display some of these in section 2.2 of the S1 Appendix (Fig H through Fig O). We ran this process with the SIR model for each new level of missing data, recalculation of centrality (or relevant measure), and immunized target nodes, each time recording the size of the SIR outbreak. The means are then compared between strategies at given levels of missing data to determine their relative effectiveness. All comparisons discussed in the results were verified with t-tests using the distributions of means (full significance tests in section 2.1 of the S1 Appendix, Fig E through Fig G).

## Results

### Simulations on the Colorado Springs network

Our first set of results is based on the synthetic "medium transitivity" Colorado Springs network. We found unsurprisingly that random immunization was the worst strategy at every level of missing data and immunization coverage (see Fig 2). All targeted strategies reduce outbreak size further than random immunization, despite random immunization being perfectly robust to missing data (as a random choice of nodes to immunize from a random sample of nodes is still perfectly random). This is an important finding: even at extremely high levels of missing nodes (e.g., 80% missing), targeted immunization was still preferable to random

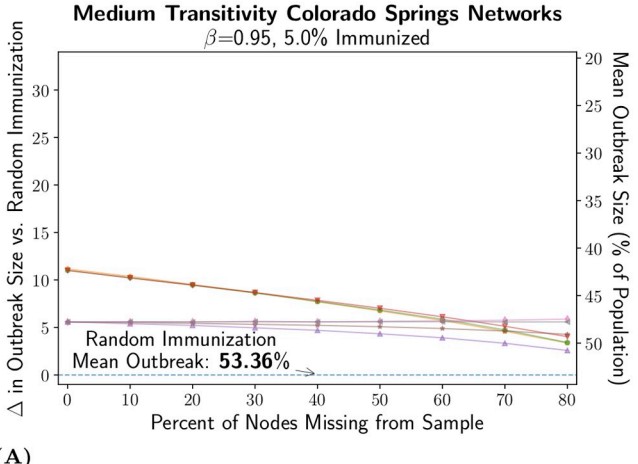

**(A)**

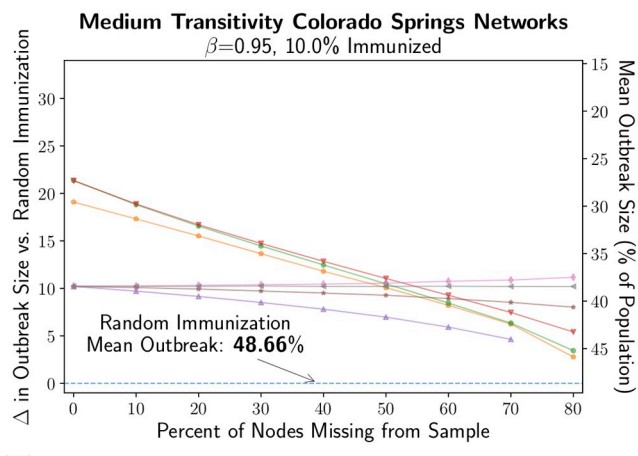

**(B)**

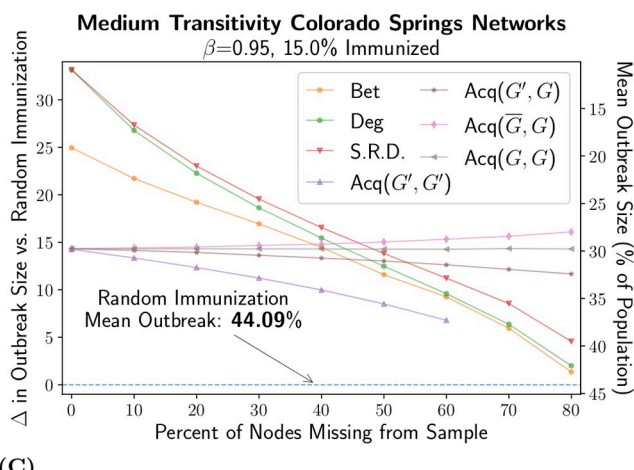

**(C)**

**Fig 2. Effectiveness of Immunization Strategies at Different Levels of Missing Data and Immunization Coverage.**
Data points represent the difference in the mean final outbreak size of random immunization and each targeted
immunization strategy (in percent of the population). Error bars were omitted because marker size was larger than
95% confidence intervals on the mean (5000 individual networks with 2000 SIR simulations for each). With $\beta = 0.95$,
this ensemble of networks had a measured $R_0$ value of 4.85 (computationally assessed [76]). When no immunization
was applied, the mean outbreak size was 58.12% of the population. Data points were omitted if a strategy was ever
unable to converge to that level of immunization coverage, i.e., when there were not enough candidates to immunize.
The legend lists all strategies tested: betweenness (Bet), degree (Deg), self-reported degree (S.R.D), and all variants of
acquaintance immunization (Acq($G'$, $G'$), Acq($G'$, $G$), Acq($\overline{G}$, $G$), and Acq($G$, $G$)).

immunization by a considerable margin. In fact, random immunization had significantly higher outbreak levels than every other strategy, under every possible condition.

Degree, self-reported degree, and betweenness immunization all outperformed random immunization, but also showed deteriorating effectiveness as missing data levels increased. These strategies were effective (i.e., low outbreak size) when missing data were low, with worse performance as missing data increased (see any panel in Fig 2). For example, at 5% immunization, the expected outbreak size under degree immunization went from around 43% with no missing data to 50% with 80% missing; with 15% immunization, the outbreak size increased from 11% to 42%. This also meant that the benefit of increasing immunization coverage (e.g., moving from 5% to 15% coverage) was higher when missing data were low.

The global strategies of betweenness, degree and self-reported degree outperformed the stochastic strategies, based on acquaintance immunization, only at low levels of missing data. Indeed, the global strategies were initially superior relative to acquaintance immunization, but rapidly became less effective as missing data increased, whereas $(G', G)$ lost its effectiveness more slowly, the $(G, G)$ lines were flat and $(\bar{G}, G)$ strategy actually improved slightly. The crossover point differed across the different values of immunization coverage, but all fell between 40% and 80% missing data. As the proportion of immunization coverage was reduced, the stochastic measures did not surpass the global strategies until higher levels of missing data were reached since the effect of missing data on the global strategies was weaker. In general, however, we saw that the relative ranking of the strategies was quite consistent at the highest and lowest levels of missing data.

Fig 2 also highlights a more surprising result: $(\bar{G}, G)$, which starts with an initially incomplete sampling frame but updates it when acquaintance immunization leads to the discovery of new nodes, was as effective or better than $(G, G)$, the equivalent strategy with complete information. This result held even at extremely high levels of missing data and low levels of immunizations. In fact, the $(\bar{G}, G)$ variant was found to actually increase in effectiveness as levels of missing data increase. This most likely stems from the fact that, by coupling data discovery with acquaintance immunization, $(\bar{G}, G)$ discovers important nodes that were initially missed while ignoring lower degree nodes. Indeed, if the initial data collection missed a central node of high degree, the acquaintance immunization is likely to find that node even when starting with an incomplete sampling frame. Therefore, while an updating $(\bar{G}, G)$ with 5% level of immunization is unlikely to fully recover an 80% level of missing data, it will recover the most important pieces of missing data. For example, with the medium transitivity networks at 80% missing data, we observed that by the point in the $(\bar{G}, G)$ acquaintance process where the desired 10% of nodes to target had been identified, the sample of nodes had become positively biased towards high degree nodes, with a mean degree of 9.7 compared to the overall mean degree of the network of 8.46.

The results for the low and high transitivity networks, presented in Fig 3, were generally consistent with the medium transitivity results. One key difference was that $(\bar{G}, G)$ did not improve with missing data on networks with low transitivity, suggesting the relevance of transitivity and/or outbreak size for targeted data recovery. Still, all targeted strategies outperformed random immunization, while degree and betweenness were the optimal strategies up to a threshold of missing data between 40% and 80%, where acquaintance immunization, $(G', G)$, $(G, G)$, or $(\bar{G}, G)$, became the best choice.

Altogether, the results so far summarize two of our main findings. First, there exists a threshold of missing data over which local stochastic strategies like acquaintance immunization outperformed global strategies based on network centrality metrics. Second, acquaintance immunization strategies which discover missing nodes adaptively are just as effective or better than equivalent strategies with complete data.

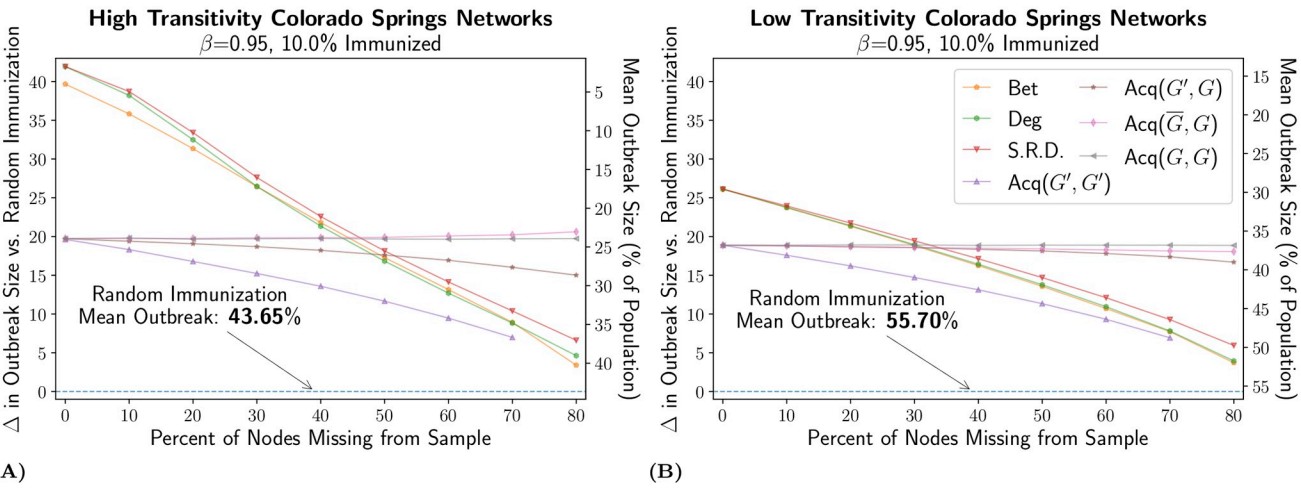

**Fig 3. Effectiveness on High and Low Transitivity Networks.** We use the same visualization scheme as in Fig 2 and show results for the medium immunization coverage of 10% on networks based on the Colorado Springs data with high and low transitivity. For the value of $\beta = 0.95$ used in these tests: the ensemble of high transitivity networks (A) had a measured $R_0$ value of 4.64 and a mean outbreak size of 52.58% of the population with no immunization; the ensemble of low transitivity networks (B) had a measured $R_0$ value of 6.98 and a mean outbreak size of 65.87% of the population with no immunization. The legend lists all strategies tested: betweenness (Bet), degree (Deg), self-reported degree (S.R.D), and all variants of acquaintance immunization ($\text{Acq}(G', G')$, $\text{Acq}(G', G)$, $\text{Acq}(\overline{G}, G)$, and $\text{Acq}(G, G)$).

## Additional analyses

Our first set of additional analyses studied the low, medium, and high transitivity variants of the Colorado Springs network over a wide range of input parameters. We systematically varied the level of transmissibility in the simulation, allowing $\beta$ to range from 0.2 to 1.7, a range of values higher and lower than that used in the main results (0.95). These $\beta$ values lead to $R_0$ values in line with infections as diverse as hepatitis C to mumps. The results, presented fully in section 2.3 of the S1 Appendix, suggest that the level of infectiousness does not dramatically affect the relative ranking of the immunization strategies.

The second set of additional analyses extends the results using the Add Health network of friendships between adolescents in school. The analysis allows us to generalize the results to networks with very different structural features, and where different kinds of pathogens (such as mononucleosis) might spread. As shown in Fig 1C, the Add Health networks lack a long tail for the degree distribution; unlike the Colorado Springs networks, a few actors do not disproportionately determine the structure of the school networks.

Results using the Add Health network are shown in Fig 4 and feature some important differences from our previous results. The targeted immunization strategies still all improved on the performance of random immunization, but here the reduction in outbreak size was smaller than in the Colorado Springs networks. For example, in the Add Health network with 10% immunized, 40% missing and $\beta$ set to 0.95, degree centrality yielded a 5% lower final outbreak size than random immunization; compare this to a 13% difference in the Colorado Springs network (medium transitivity). This is the case, in part, because the Add Health network has few (or no) extremely central nodes. This means that the most central nodes have only marginally higher centrality than less central nodes. Thus it can be harder to select who should be immunized (especially with missing data), reducing the returns to targeted immunization in such cases.

The structure of the Add Health network also affected some of the specific findings related to $(\overline{G}, G)$. In the Add Health network, the $(\overline{G}, G)$ strategy sometimes performed worse than

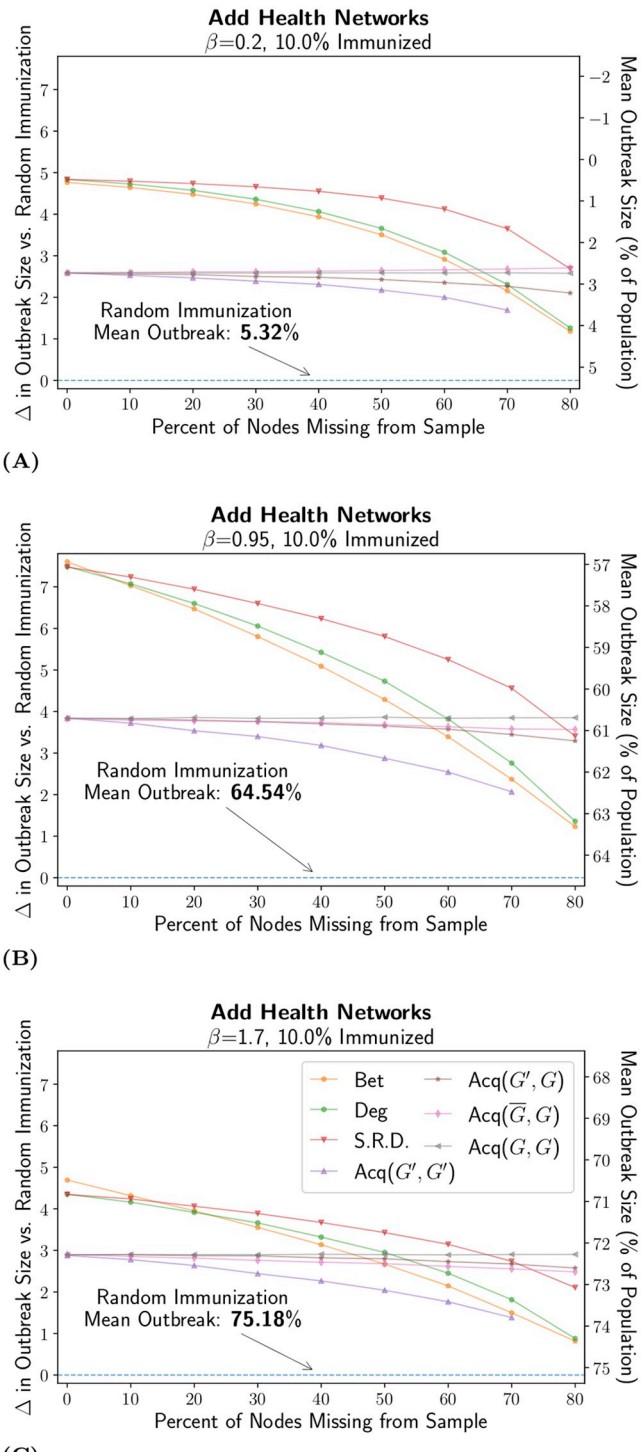

**Fig 4. Effectiveness on Add Health Networks over a Range of Transmission Rates.** We use the same visualization scheme as in Fig 2 and the medium immunization coverage of 10% on networks based on a typical school network from the Add Health study. We varied $\beta$ to produce both small and large outbreak sizes. For panel (A), the measured value of $R_0$ was 1.40, and the mean outbreak size with no immunization was 11.12% of the population. For panel (B), the measured value of $R_0$ was 3.34, and the mean outbreak size with no immunization was 75.68%. For panel (C), the measured value of $R_0$ was 3.84, and the mean outbreak size with no immunization was 86.23%. The legend lists all strategies tested: betweenness (Bet), degree (Deg), self-reported degree (S.R.D), and all variants of acquaintance immunization (Acq($G'$, $G'$), Acq($G'$, $G$), Acq($\overline{G}$, $G$), and Acq($G$, $G$)).

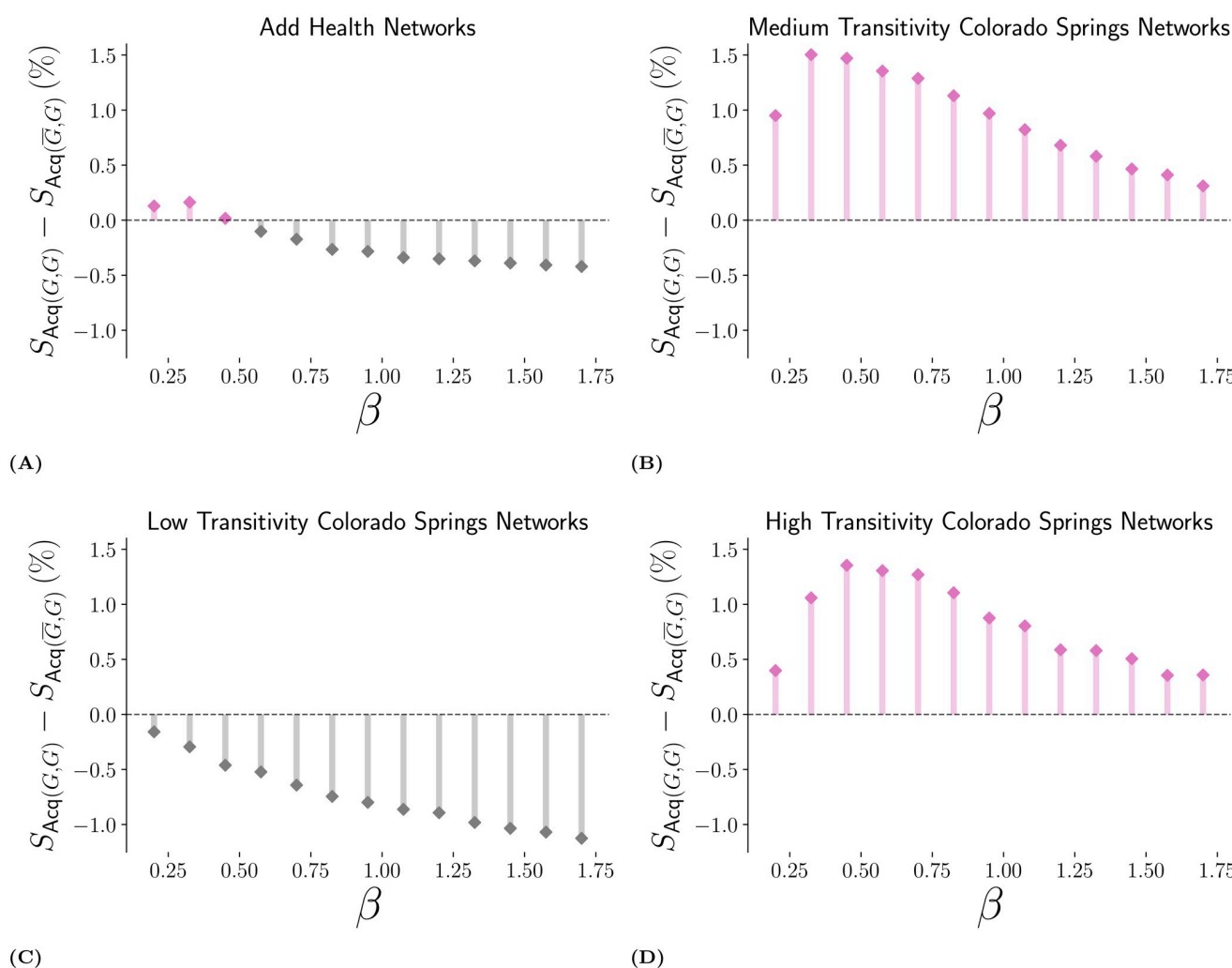

**Fig 5. Comparative Effectiveness of the Updating-sample Strategy (Acq($\bar{G}$, $G$)) Versus the Strategy with Access to Complete Census (Acq($G$, $G$)).**
Examined in terms of the difference in their outbreak size (both measured in percent of network infected). This comparison is assessed at the highest level of missing data, over a range of $\beta$ to see how this comparative effectiveness varies with the regime of transmission probability. The immunization level for all tests shown here is 10%. In panel (A), the measured value of $R_0$ ranged from 1.40 to 3.84. In panel (B), the measured values for $R_0$ ranged from 2.91 to 5.15. In panel (C), the measured values for $R_0$ ranged from 3.49 to 7.79. In panel (D), the measured values for $R_0$ ranged from 2.81 to 4.95.

($G$, $G$), specifically at higher levels of infectiousness. This observation again suggests that low infectiousness might be a key criteria for the targeted ($\bar{G}$, $G$) to outperform the complete information of ($G$, $G$), but also stresses the importance of central hubs and degree heterogeneity.

Finally, an extensive comparison of the relative performance of ($\bar{G}$, $G$) and ($G$, $G$) under different values of infectiousness and under different network structures is shown in Fig 5. These results confirmed our previous observations: The targeted data recovery scheme implemented by ($\bar{G}$, $G$) performed better when infectiousness was low and the network structure featured significant degree heterogeneity. Most likely, this stems from the fact that targeted data recovery will find central nodes if those are also high-degree nodes, and that weak epidemics will be concentrated around these central hubs. When facing an outbreak close to its epidemic threshold on heterogeneous networks, we therefore expect that an intervention using an incomplete sampling frame and the targeted data recovery scheme of ($\bar{G}$, $G$) would outperform an equivalent intervention using a known and complete census. That being said,

at extremely low transmission rate ($\beta \to 0$) both strategies must be equivalent since outbreaks are simply impossible.

To summarize our results, the rank ordering of the strategies at different levels of missing data was found to be mostly independent of network structure and transmission rate. Similarly, acquaintance immunization strategies were found to be significantly more robust to missing data than the typical benchmark strategies requiring global information. In particular, acquaintance immunization that leverage the intervention for targeted data recovery were found to improve with higher levels of missing data, unlike any other approaches.

## Discussion

In this paper, we have produced and implemented a method of comparing both global and stochastic targeted immunization strategies under conditions of imperfect data. Nearly two decades of research suggest that these types of strategies would be of considerable help in reducing the spread of contagion. Our research represents an important step in reconciling past research with the common reality of imperfect network census coverage—introducing a method to formally examine the intuitive notion that strategies which perform well under perfect conditions may be ineffective when there is missing data. While missing data is ubiquitous in network science, this reconciliation is especially important in contexts where data collection is difficult, such as when dealing with vulnerable hidden populations, networks based on unobtrusive observation, and in projects with limited resources.

Our most consistent finding is the clear superiority of targeted immunization over random immunization, even at very high levels of missing data (e.g., 80% missing nodes). This suggests that targeted immunization strategies are useful even in cases where the prospects for complete network data are relatively poor—a positive sign for the applicability of targeted immunization strategies.

We also find that missing data generally leads to higher outbreak levels, with some strategies more affected by missing data than others. For example, betweenness and both degree strategies perform best at low levels of missing data, but tend to converge with other strategies as missing data increases. Additionally, at very high levels of missing data, our three variants of acquaintance immunization—$(G', G)$, $(G, G)$ and $(\bar{G}, G)$—outstrip both degree strategies and betweenness immunization and have the lowest outbreak sizes relative to other strategies.

Importantly, the variant $(\bar{G}, G)$ which couples immunization with additional data collection, often performs as well or better than acquaintance immunization with complete data $(G, G)$, while relying on nodes in the incomplete sampling frame. This important result stems from the fact that the targeted immunization procedure is leveraged by $(\bar{G}, G)$ as a targeted data recovery scheme. Moreover, this strategy was the only one that was able to perform better with higher levels of missing data. This result opens up a promising new avenue of research for targeted data recovery in networks with missing data, an idea which could be powerful even beyond immunization problems. And at the very least, this idea stresses the importance of at least acknowledging that the available data might be incomplete.

Overall, while some strategies are more sensitive to missing data, they still may be an optimal choice over more robust strategies. This is the case as the less robust strategies tended to be more effective when missing data were low. We claim the following. First, the relative effectiveness of an immunization strategy with full network knowledge does not accurately predict its relative effectiveness with incomplete data. Second, immunization strategies which do not need global data may still be affected by incomplete sampling frames and local data. Third, even when stochastic strategies use perfect information, a less robust strategy, like betweenness, may be more effective at many levels of missing data and immunization coverage. Finally,

the relative effectiveness of a robust strategy compared to a non-robust strategy is context specific. The choice is rarely so simple as global versus stochastic, with the optimal strategy depending heavily on the level of missing data and the immunization coverage.

Our results are based on a wide range of scenarios, but we hope to see future work extend our analysis, considering other immunization strategies, different network topologies, different outbreak models, coevolution of outbreaks and interventions, missing data related to the infection status of participants, and other different types of missing data including non-random missing nodes and edges [32]. Our method is designed with versatility in mind, and can be easily generalized to have the complete portion of the data originate from any sampling process. It is also easy to accommodate a wide variety of biases and other complications. This allows our method to address the larger problem of immunization of partially-observed networks. Betweenness centrality, for example, is difficult to measure accurately when the network is small and the missing nodes are central to the network [78], and may prove ineffective as an immunization strategy under such conditions. Future work should thus consider the effect of missing central nodes on the effectiveness of different immunization strategies.

There are a number of practical implications for our study. Researchers "in the field" should, ideally, pay attention to both the effectiveness and robustness of the proposed immunization strategy. A researcher should not necessarily avoid a strategy that is affected by missing data (like degree or betweenness) as it still may be the best overall choice, better than more robust strategies. While our model still involves a number of simplifications, a researcher could use our results as a rough guide to consider this tradeoff. Armed with local knowledge of their particular population, sampling methods, and immunization logistics, including how the contagion spreads, the nature of the missing data, and the nature of what constitutes an immunization, one could adapt our methodology with more specific models for the outbreak (e.g. Refs. [79, 80]), for the missing data (e.g. types [32, 48, 81] or estimates [45, 82, 83]), and for an immunization campaign (e.g. incorporating imperfect immunization and ongoing immunization during an epidemic) to determine an optimized subset of nodes to immunize. The hope, more generally, is that a team designing an intervention in a real-world scenario could use our results (and similar/future work) to inform both the data collection and intervention campaign. To this end we have made the code used to produce these simulations publicly available.

## Supporting information

**S1 Appendix. Supporting information. Additional analyses and figures to support the results of the paper**.
(PDF)

## Acknowledgments

The authors acknowledge the Vermont Advanced Computing Core at the University of Vermont for providing High Performance Computing resources, Guillaume St-Onge for providing the simulation software as well as Andrew Becker and Jane Adams for their useful comments. The primary author would also like to thank a number of mentors for early guidance in this project including Patrick Habecker, Elspeth Ready, Kirk Dombrowski, Alan Jamieson, and Andrew Cognard-Black.

## Author Contributions

**Conceptualization:** Samuel F. Rosenblatt, Jeffrey A. Smith, G. Robin Gauthier, Laurent Hébert-Dufresne.

**Data curation:** Samuel F. Rosenblatt, Jeffrey A. Smith.

**Formal analysis:** Samuel F. Rosenblatt, Jeffrey A. Smith, Laurent Hébert-Dufresne.

**Funding acquisition:** Jeffrey A. Smith, G. Robin Gauthier, Laurent Hébert-Dufresne.

**Investigation:** Samuel F. Rosenblatt, Jeffrey A. Smith, Laurent Hébert-Dufresne.

**Methodology:** Samuel F. Rosenblatt, Jeffrey A. Smith, G. Robin Gauthier, Laurent Hébert-Dufresne.

**Project administration:** Jeffrey A. Smith, G. Robin Gauthier, Laurent Hébert-Dufresne.

**Resources:** Jeffrey A. Smith, Laurent Hébert-Dufresne.

**Software:** Samuel F. Rosenblatt, Jeffrey A. Smith.

**Supervision:** Jeffrey A. Smith, G. Robin Gauthier, Laurent Hébert-Dufresne.

**Validation:** Jeffrey A. Smith, G. Robin Gauthier, Laurent Hébert-Dufresne.

**Visualization:** Samuel F. Rosenblatt, Laurent Hébert-Dufresne.

**Writing – original draft:** Samuel F. Rosenblatt, Jeffrey A. Smith, G. Robin Gauthier, Laurent Hébert-Dufresne.

**Writing – review & editing:** Jeffrey A. Smith, G. Robin Gauthier, Laurent Hébert-Dufresne.

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
