## [Decision Letter · Decision Letter 0]

18 Dec 2019

Dear Dr Hebert-Dufresne,

Thank you very much for submitting your manuscript 'Immunization Strategies in Networks with Missing Data' for review by PLOS Computational Biology. Your manuscript has been fully evaluated by the PLOS Computational Biology editorial team and in this case also by two independent peer reviewers. The reviewers appreciated the attention to an important problem and the derivation of potentially interesting findings, but raised some substantial concerns about the manuscript as it currently stands. While your manuscript cannot be accepted in its present form, we are willing to consider a revised version in which the issues raised by the reviewers have been adequately addressed. We cannot, of course, promise publication at that time.

Sincerely,

Corina E Tarnita

Associate Editor

PLOS Computational Biology

Rob De Boer

Deputy Editor

PLOS Computational Biology

[LINK]

Reviewer's Responses to Questions

**Comments to the Authors:**

Reviewer #1: This manuscript examines the spread of contagious elements in a network population. This is a simulation based study in which the authors examine eight different immunization strategies over a network with variable vaccination rate, transitivity, and information on the network itself. They then apply an epidemic model over many stochastic realizations to assess mean final epidemic size. They find tradeoffs, particularly in acquaintance immunization between information required and robustness to missing data. These results could be applicable to a number of epidemic settings.

Between the main text and supplement, this is a very dense manuscript. This is somewhat the manuscript’s weakness as there is a good amount of notation, while the manuscript strives to provide a general approach to contagion phenomena. However, there were multiple times in the manuscript where I was uncertain what exactly was being modeled and, therefore, the key implications of the results. My main reservations are based on how the epidemic model was formulated over the network, and what approximate disease and population were being examined. Honing in on modeling a specific setting would strengthen the manuscript.

Major points:

1. In section 2.4 and beyond, it was unclear to me what situation / disease was being modeled. There is a substantial discussion about HIV spread in section 2.1, and the networks utilized seem based on that, so I assumed the disease being modeled was HIV. However, a Susceptible-Infected-Recovered model was used. Disease-specific time units aren’t mentioned here, but if one supposes they are in days the infectious period is 5 days which would indicate more of a childhood disease or a flu-type pathogen. The authors do utilize a range of parameters, but they are not so drastically different. Depending on this, it’s unclear how appropriate the chosen base network is. For example, childhood diseases will require more information on classroom, household location and sibling, age, and school (see Hagelloch data and manuscripts from Groendyke & Welch). These networks will certainly be different given children vs adults mixing patterns. Without knowing how R0 was calculated, it was unclear how transmissible a pathogen was being discussed, which was then hard to reconcile with the vaccination levels of 2.5% to 10%. For some diseases / populations / eras this could be possible, but given something like a flu or measles, we may expect these to be much higher, up to 50% or even > 90%. I also wasn’t sure if these vaccination levels were sustained throughout the epidemic duration. Lastly, I assumed vaccinated nodes were moved to the ‘Recovered’ class of the model, but then is everyone assume to be susceptible at the start of the epidemic? Overall in this section I had trouble following what specific pathogen / population was being modeled and it was thus hard for me to interpret the results of epidemic final size.

2. I understand plotting constraints given the many scenarios, but it seems necessary to add some measure of variability to these final size estimates given the stochastic nature of network epidemic models. From a practical point of view, this is important as if a particular strategy leads to a particularly long distribution tail, e.g., “super-spreaders” a la Lloyd-Smith, 2005, that may be an undesirable result for those planning an intervention strategy. A relatively simple summary static such as fitting negative binomial distributions to the epidemic final sizes would be beneficial as a result. The caption for Figure 2 does mention that 95% CIs were smaller than the marker size which, given the stochastic epidemic model on a network, seems slightly unusual.

3. The problem of missing data seems two-fold. There is the case where information on particular nodes in the network are missing, which is discussed here. But there is also the case of missing epidemic information. This may be incorporated into missing node information itself, but it seems there could be a very likely scenario where you have information on node X in terms of their contacts and connectivity, etc. However, in a real world setting you may not necessarily know if X gets infected and then spreads. Could this be incorporated into this model? I’m not necessarily suggesting to do that here, as there are many results discussed, but a discussion, and more clear description of the assumptions (e.g., 100% reporting), of this topic in the discussion section may help target the manuscript to the type of epidemic scenarios for which these results would be most applicable.

Minor points:

Line 23: Unclear phrasing at the start of the sentence: pathogens are not inherently negative.

Line 33: Immunization maybe should be changed to intervention if reduction of contact is included

Line 216: Unclear what ‘attempted to saturate the population’ means

Line 231-238: It would be helpful to give a real world example of what a low transitivity (0.01) network would represent

Line 246: Does clustering mean the same as transitivity here? If so I would use consistent language

Line 383: How was vaccination implemented into the SIR model? Is there vaccination during the course of the epidemic? If so, is it targeted? If not, it may be worthwhile to mention that in the discussion section as a limitation.

Line 383: The actual model uses immunization as vaccination, but the language in the introduction states that immunization is any intervention. This language should be standardized throughout the manuscript

Line 395-399: What are the time units here? What is the R0 of the pathogen, and how is it calculated?

Description of the SIR model: when does the epidemic end?

Figure 2 caption: What does it mean for a strategy to not be able to converge to an immunization level?

Figure 2: What was the epidemic size in the absence of any intervention strategy? If it was for example 90% of the population, then the random intervention may be still somewhat impressive. But again this depends on what disease is being modeling here.

Final paragraph for the discussion: will code be made available to adapt this analysis to other situations, networks, and models?

Reviewer #2: This paper examines the optimal way of allocating vaccinations among a population when the transmission network over which the disease spreads is known, but with error. A large body of previous work over the past 20+ years has focused on optimal vaccination strategies when resources are limited, but has always assumed that everything about the network is known. In parallel, a large body of work from network sciences has focused on estimating network properties in an unbiased way when there is missing data and different biases in sampling strategies. This paper combines these two ideas to understand what network-based vaccine distribution schemes are optimal when networks are imperfectly sampled.

Overall I think this is a really nice addition to the field. The authors clearly show that some network-based vaccination strategies are still way better than random allocation strategies even under realistically high levels of missing data. They show how the optimal strategy depends on some properties of the network, the nature of the missing data, and the details of how the network is used to allocate the vaccine. Their methods are easy to follow and the introduction is a very thorough review of the field. This paper is an important step in connecting the mainly theoretical work done on vaccine allocation with the real-world constraints of mapping out risk networks for real diseases.

I have a few specific comments that are included below. Overall, my more major concerns are the following:

1. Some parts of the paper are really wordy and it takes a really long time to get through. This includes the introduction (Section 1) and the Materials & Methods (Section 2). And the Results section reads more like a Figure caption than an appealing narrative of some very nice findings! I think that all of this could be a lot more concise and reader-friendly with some careful editing.

2. The results section of the paper is pretty small compared to the lead up. I don’t think the conclusions are adequately explored for their relevance in slightly different scenarios. I think that the authors could easily explore a few more scenarios to fill out this section to the size normally expected in a PLoS Computational Biology paper. For example, why not explore some other network structures that are very different from the Colorado Springs network? Why not consider some other infection models, like SIS? These are mentioned in the Discussion as future work but would be more appropriately included in this manuscript

More specific (minor) comments:

* The Author Summary is just as technical as the Abstract - not appropriate for a lay scientific audience

* Introduction:

* This section is VERY long

* Line 25 - Papers from year 2000 would not be characterized as “recent work” by most people

* Most of these citations are to extremely theoretical papers by physicists. Do you have any examples of people in the field of infectious disease epidemiology considering network-based immunization strategies? Can you convince the reader that there any real world practicality to such ideas? I’m not sure just citing papers of immunizing healthcare workers counts. This is an entire group of people, and doesn’t really use any specific concept of networks.

* Materials & Methods

* Overall this section is very wordy. Could be shorted by just writing more concisesly, and with some details moved to figure captions

* Figure 1. Caption needs more details. Does a degree here correspond to either a close friendship, sexual contact, or drug co-use, or just one of these relationships? Also there is a big enough population size that it would be clearer to have each degree as its own bin instead of two degrees per bin, so it would be clear how many individuals have zero vs one vs two degree, etc. It would also be nice to show an image of the graph structure. Degree distribution doesn’t fully capture the structure of a network, like how much clustering there is.

* It would be also worth mentioning re the Colorado Springs network that HIV is only one of the many infections that is commonly spread by persons with infection drug use; Hep A, B, and C, as well as other STDs, are also common in these populations

* And PWID stands for “PERSONS who inject drugs”

* Line 292 - Degree immunization - It’s not totally clear what you mean when you say “selects nodes based on network degree”. Do you rank based on network degree then choose the top X? What if there are ties? Or is selection random but proportional to degree? Directly proportional to degree or on a log or other scale? Same question for some of the other strategies

* Section 2.3: Are there ever scenarios in which edges are missing at random, even though the nodes they may be connecting are in the network?

*

Nice descriptions of the different immunization strategies

* Do you also consider immunization under FCD and RDS, as the previous papers you cited around line 170 did? Just to repeat their results and compare to yours to be as comprehensive as possible?

* Section 2.4: I’m a bit confused by the description of the SIR model. First, you say individuals "pass the infection to all of their nearest neighbors in the network”. However, that is not a correct way to simulate a stochastic infection model. Each infected node should have a constant probability (gamma) of recovery per unit time, and each susceptible node should have a probability dependent on its number of infected neighbors (beta*n_infected) of becoming infected. The exponentiated equations the authors write involving time period Delta only hold if time Delta is very small, such that the probability that more than one event happening in the same time interval is negligible (e.g. there is no chance that an infected neighbor will recover before it infects you)

* Results

* This section reads like a figure caption. It would be much clearer to describe the FINDINGS, not narrate the graphics (e.g. what is on the x vs y axes)

* Figure 2 caption: Include explanation of abbreviations in legend, e.g. S.R.D = Self-reported degree

* Figure 2/3/4 all look really similar. Is there not a more creative way to show some of these trends? For example, if you want to show the effect of changing transitivity, make a graph with transitivity as the x axis. Same could be said for % immunized. I personally would find it more intuitive if the y axis on all the graphs was not final infection size but the opposite, e.g. % not infected, so that higher values would be “better”. It seems like the main trend of interest to someone designing a vaccination strategy is % reduction in infection vs % immunized, and how this trend depends on details of missing data, immunization strategy, and network structure, so it might be more intuitive to present the data this way.

* I think there are many times when you cite Figure 3 but you really mean to cite Figure 2.

* I found the results section rambling and hard to follow. Instead of starting each paragraph with “Figure X shows …”, start with a topic sentence summarizing a finding you observed, e.g. “Random immunization was the worst strategy at every level of immunization coverage and missing data”. If the finding is seen in a particular figure, add it as a reference, e.g. “(Fig X)”.

* Discussion

* Overall, very nice

* Line 609: I don’t think you have defined yet what you mean by robust vs efficacious

**Have all data underlying the figures and results presented in the manuscript been provided?**

Reviewer #1: No: I could not find an attachment or link to the data.

Reviewer #2: No: This is a modeling paper, so not really much data. However data for the Colorado Springs network used should be provided.

PLOS authors have the option to publish the peer review history of their article (what does this mean?). If published, this will include your full peer review and any attached files.

Reviewer #1: No

Reviewer #2: No

---

## [Decision Letter · Decision Letter 1]

22 Apr 2020

Dear Dr. Hebert-Dufresne,

We are pleased to inform you that your manuscript 'Immunization Strategies in Networks with Missing Data' has been provisionally accepted for publication in PLOS Computational Biology. Please do note, however, that both reviewers had a few minor suggestions and recommendations. Although acceptance does not hinge on your implementation of these suggestions, we encourage you to consider them carefully.

Best regards,

Corina E. Tarnita

Associate Editor

PLOS Computational Biology

Rob De Boer

Deputy Editor

PLOS Computational Biology

Reviewer's Responses to Questions

**Comments to the Authors:**

Reviewer #1: This manuscript examines the spread of contagious elements in a network population. This is a simulation-based study in which the authors examine eight different immunization strategies over a network with variable vaccination rate, transitivity, and information on the network itself. They then apply an epidemic model over many stochastic realizations to assess mean final epidemic size. They find tradeoffs, particularly in acquaintance immunization between information required and robustness to missing data.

This revision contains a substantial reworking of the text as well as additional analyses. The supplement is very complete with multiple transmission and network scenarios. In particular, the focus of the manuscript has been made much clearer and the Intro, Methods, and Results section have been streamlined to this end. Methods section 2.4 is much clearer. The inclusion of R0 in each scenario is useful. In particular, I appreciate the addition of section 2.2 Analysis of variability of outbreak sizes in the Supplement. The 50% immunized scenario is useful as well. The very different Add Health network is a useful comparison.

Overall, all of my concerns have been addressed. I only have a few minor points, mostly issues of clarity in the introduced text, as follow:

Minor points:

Line 33 in intro: It would be helpful define “MSM” here

Line 373: The extra information on calculating R0 is helpful (both in the text and response letter), I would just add a reference to the software itself at the end of line 374.

Line 425: “were” should be “was”

Line 462: “summarizes” should probably be “summarize”

Line 470: “transmissibility” may be more accurate than “infectiousness” since beta is being varied

Line 594: I noted, and understand, that processing the new code at publication quality prior to the resubmission was a time consuming task — hopefully this still possible for the final version of the manuscript. A reference or link to the code here would help the research community further utilize these results.

Figure 1 caption: If I’m understanding correctly, the last sentence seems redundant with the first

Supp Fig 23 caption: The last sentence of this caption is unclear

Reviewer #2: The authors have addressed all of my concerns from the first round of reviews, and I believe they have adequately addressed all the other reviewer's concerns as well. The paper now looks great.

The only remaining suggestion I have is that it is strange to have a section title including the phrase "SIR diffusion model", because the SIR model is an infection model, which is not mathematically the same as a diffusion model, so this is quite confusing

**Have all data underlying the figures and results presented in the manuscript been provided?**

Reviewer #1: Yes

Reviewer #2: No: There is no new data in this paper, but authors should probably provide a link to a repository with the code used to generate the main results, as well as the network structures

PLOS authors have the option to publish the peer review history of their article (what does this mean?). If published, this will include your full peer review and any attached files.

Reviewer #1: No

Reviewer #2: No

---

## [Editor Report · Acceptance letter]

30 Jun 2020

PCOMPBIOL-D-19-01722R1 

Immunization Strategies in Networks with Missing Data

Dear Dr Hebert-Dufresne,

I am pleased to inform you that your manuscript has been formally accepted for publication in PLOS Computational Biology. Your manuscript is now with our production department and you will be notified of the publication date in due course.

With kind regards,

Laura Mallard
